# Novel Horizons in Postbiotics: *Lactobacillaceae* Extracellular Vesicles and Their Applications in Health and Disease

**DOI:** 10.3390/nu14245296

**Published:** 2022-12-13

**Authors:** Elena González-Lozano, Jorge García-García, Julio Gálvez, Laura Hidalgo-García, Alba Rodríguez-Nogales, María Elena Rodríguez-Cabezas, Manuel Sánchez

**Affiliations:** 1Department of Pharmacology, Biomedical Research Center (CIBM), University of Granada, 18012 Granada, Spain; 2Biosanitary Research Institute of Granada (ibs.GRANADA), 18012 Granada, Spain; 3CIBER Hepativ and Digestive Diseases (CIBEREHD), 28029 Madrid, Spain

**Keywords:** EVs, IBD, immunomodulation, inflammation, *Lactobacillus*, postbiotics, probiotics

## Abstract

*Lactobacillus* probiotics contained in dietary supplements or functional foods are well-known for their beneficial properties exerted on host health and diverse pathological situations. Their capacity to improve inflammatory bowel disease (IBD) and regulate the immune system is especially remarkable. Although bacteria–host interactions have been thought to occur directly, the key role that extracellular vesicles (EVs) derived from probiotics play on this point is being unveiled. EVs are lipid bilayer-enclosed particles that carry a wide range of cargo compounds and act in different signalling pathways. Notably, these EVs have been recently proposed as a safe alternative to the utilisation of live bacteria since they can avoid the possible risks that probiotics may entail in vulnerable cases such as immunocompromised patients. Therefore, this review aims to give an updated overview of the existing knowledge about EVs from different *Lactobacillus* strains, their mechanisms and effects in host health and different pathological conditions. All of the information collected suggests that EVs could be considered as potential tools for the development of future novel therapeutic approaches.

## 1. Introduction

The gastrointestinal tract (GIT) of mammals is a complex ecosystem of host cells, available nutrients, and a high and diverse number of microorganisms, which is estimated to exceed 10^14^ [1,2]. All of these microorganisms constitute the gut microbiota, and comprise bacteria, archaea, fungi, protozoa, and viruses, which cohabit and interact mutually with the host [3]. Nowadays, there is enough evidence to consider the gut microbiota as a new organ that plays a relevant role in the normal physiology of the GIT, including maintenance of the epithelial barrier, modulation and maturation of the immune system, regulation of neurotransmitters [4], degradation of nondigestible carbon sources and synthesis of short-chain fatty acids (SCFAs), and production of several metabolites, including vitamins (Figure 1) [5,6]. Conversely, the disturbance in the microbiota composition and function, known as dysbiosis, can disrupt the human intestinal homeostasis [7]. In fact, it is widely understood that gut dysbiosis is associated with many pathological conditions, including inflammatory bowel disease, obesity, diabetes, irritable bowel syndrome, allergy, and mood disorders [8,9]. Accordingly, the restoration of gut dysbiosis has been considered as a potential therapeutic approach. In this sense, the administration of probiotics is becoming a promising strategy in the treatment of different human diseases.

The Food and Agriculture Organization (FAO) and the World Health Organization (WHO) define probiotics as “live microorganisms that when administered in adequate amounts confer a health benefit on the host” [10]. This definition covers a general concept of the actions exerted by all probiotics. However, each strain has its specific characteristics that cannot be extrapolated to others (including the same genera). Consequently, it is necessary to scientifically demonstrate the efficacy and mechanisms of action of each strain independently [3]. Despite this, most probiotics display the following beneficial effects: (i) competitive exclusion of pathogenic microorganisms and secretion of antimicrobial substances, (ii) enhancement of the epithelial barrier function, (iii) production of short-chain fatty acids (SCFAs) and other metabolites, (vi) restoration of the homeostasis of the intestinal microbiota, and (v) immunomodulatory activity [11].

Considering the definition proposed by FAO/WHO, the term probiotic should be restricted to living microorganisms. However, an increasing amount of scientific evidence has revealed that the administration of inactivated or non-viable microbes can also benefit human health. Therefore, and in line with the concept of probiotics, other terms such as paraprobiotic or postbiotic have emerged. In 2011, Taverniti et al. proposed the term ‘paraprobiotic’ to indicate the use of inactivated microbial cells or cell fractions to confer a health benefit to the consumer [12]. These are obtained by the cultivation and subsequent inactivation of selected strains to make them non-viable [13,14]. Paraprobiotics can be more adequate and have more predictable effects than probiotics in some clinical cases [15]. More recently, the term postbiotics has been applied to metabolites, cell-free supernatants, and soluble factors (metabolism products) secreted by live bacteria [16]. The most important postbiotics are the organic acids SCFAs, including acetate, propionate, and butyrate, as well as bacteriocins and tryptophan [17]. Postbiotics may produce beneficial effects mainly related to their anti-inflammatory, anti-pathogenic and antioxidant properties [18]. However, and depending on the type of microorganism, the strain and the metabolism product, the effects of postbiotics can vary significantly. In this context, special attention should be given to the extracellular vesicles (EVs), which have recently been reported as postbiotics with an interesting potential profile in health and disease.

Extracellular vesicles

EVs are membrane-derived lipid bilayers generated by a process known as vesiculogenesis [19]. They are capable of packaging cytosolic compounds produced by microbial metabolism, such as proteins, nucleic acids, and polysaccharides [20]. It is well-known that Bacteria, Eukarya, and Archaea can release EVs [19], which, regardless of the source, are similar in size and general composition [21]. EVs were firstly identified from the bacterium Escherichia coli using electron microscopy and, since then, EVs have been extensively studied. Among them, most of the attention has been focused on outer membrane vesicles (OMVs) from Gram-negative bacteria, due to their involvement in the virulence capacity attributed to these bacteria [22]. For example, *E. coli*, *Pseudomonas aeruginosa*, and *Actinomyces* generate toxin-containing OMVs that produce inflammatory effects similar to those produced by the entire bacteria [23,24,25]. It was not until the last decade of the 20th century that EVs from Gram-positive bacteria, named membrane-vesicles (MVs), were discovered [26,27]. In fact, it was believed that this bacterial group could not generate EVs due to their characteristic thick cell wall [28]. Therefore, both groups of bacteria, Gram-positive and Gram-negative can produce vesicles, with OMVs being generally bigger (20–200 nm in diameter) [28] than MVs (20–100 nm) [29]. Moreover, and interestingly, the structural differences between both types of bacteria imply that the EVs of each group also have dissimilar compositions (Figure 2) [29].

Several procedures have been reported to obtain EVs from bacteria. All of them include these general steps: (i) bacterial cell cultivation in adequate media, (ii) removal of cells and large cellular debris by low-speed centrifugation and sterile filtration, (iii) concentration of EVs by ultracentrifugation of filtered supernatant and purification by gel filtration or density gradient [11,30].

It has been proposed that these postbiotics, MVs and OMVs, have shown comparable benefits to the whole probiotics [31]. Remarkably, in many cases, the administration of MVs have been proposed as safer and more efficient than probiotics, avoiding the potential risks that are associated with the utilisation of live bacteria [21], specifically in critical patients with severe acute pancreatitis [32] in whom an increased risk of mortality has been associated with the use of a combination of bacteria employed as probiotic prophylaxis.

This review assesses the current knowledge about the role of MVs derived from the most important probiotic strains of the genus *Lactobacillus* in the therapeutic management of different diseases, providing significant examples of clinical studies on MVs applications and discussing the possible mechanisms involved in these effects.

## 2. *Lactobacillus*

*Lactobacillus* is taxonomically classified in the phylum *Bacillota* (synonym *Firmicutes*), class *Bacilli*, order *Lactobacillales* and family *Lactobacillaceae*. It is the most studied genus belonging to the lactic acid bacteria (LAB) group and one of the most representative groups of probiotics [33]. It has been reclassified into 25 genera [34] that include more than 240 species [35] of facultative, anaerobic, catalase-negative, Gram-positive, and non-spore-forming rods [36]. Traditionally, *Lactobacillus* species may be divided into three groups based on their metabolism: (i) obligately homofermentative group; (ii) facultatively heterofermentative group, and (iii) obligately heterofermentative group. The obligately homofermentative species ferment carbohydrates to produce lactic acid as the main by-product (e.g., *Lactobacillus acidophilus* and *Lactobacillus salivarius*), whereas the facultatively heterofermentative species, under certain conditions or with certain substrates, ferment carbohydrates to produce lactic acid, ethanol/acetic acid, and carbon dioxide as by-products (e.g., *Lacticaseibacillus casei* and *Lactiplantibacillus plantarum*). Obligately heterofermentative species ferment carbohydrates to produce lactic acid, ethanol/acetic acid, and carbon dioxide as by-products (e.g., *Limosilactobacilllus reuteri* and *Limosilactobacillus fermentum*) [37].

*Lactobacillaceae* family can be found at different localizations in the human body, including the GIT, as well as the urinary and genital systems. The most common isolates from *Lactobacillaceae* inhabiting the GIT include *Lc. casei*, *Lact. plantarum*, *Li. fermentum* and *Lacticaseibacillus rhamnosus*, *Limosilactobacillus antri*, *Limosilactobacillus gastricus*, *Lactobacillus kalixensis*, *Li. reuteri* or *Lactobacillus ultunensis* [38]. On the contrary, *Lactobacillus crispatus*, *Lactobacillus gasseri*, *Lactobacillus jensenii*, *Limosillactobacillus. vaginalis* or *Lactobacillus iners* are found more frequently in the vagina than in GIT mucosa [39]. In fact, it is well-known that the large and stable proportion of *Lactobacillaceae* genus protects the healthy female urogenital tract against pathogenic infections [39,40]. Of note, initial colonisation of neonatal skin is suspected to occur during delivery; thus, when the infants born vaginally, *Lactobacillus*, *Prevotella* or *Sneathia* species are transferred to the skin during passage through the cervix and vagina, but these species disappear by 6 weeks of age, when the microbiota begins to develop a more skin-like profile enriched with species from the genera *Staphylococcus* and *Corynebacterium* [41].

The amounts of lactobacilli found in the GIT vary depending on the age of the host and the specific location. In this context, in the adult faeces, lactobacilli bacteria account for only 0.01 to 0.06% (10^5^ to 10^8^ CFU/g) of all bacterial species, being the predominant indigenous species *L. gasseri*, *Li. reuteri*, *L. crispatus*, *L. salivarius* and *L. ruminis*. As compared with the adult microbiota, the infant faecal microbiota is more unstable and variable, ranging from 10^5^ CFU/g in the faeces of neonates, to 10^6^–10^8^ CFU/g in the faeces of infants aged >1 month. *Lp. plantarum*, *L. salivarius*, *Lc. rhamnosus*, *Lacticaseibacillus paracasei*, *Li. fermentum*, *L. gasseri*, *L. delbrueckii* and *Li. reuteri* are the most commonly isolated species in infant faeces [42].

Some of these species, such as *Lc. rhamnosus*, *Lp. plantarum*, *Lc. casei* or *Li. reuteri* have been considered as probiotics [40]. Thus, the beneficial effects attributed to *Lc. rhamnosus* or *Li. reuteri* include the management of bacterial vaginosis, atopic dermatitis, and upper respiratory tract infections, among others [43,44,45]. Similarly, *Lp. plantarum* CJLP55 supplementation showed a positive impact [46] in patients with acne vulgaris [47], whereas *Lc. casei* has been proposed as a treatment for diarrhoea in infants.

The effects observed in these studies are heterogeneous and include various pathologies. Thus, when these probiotics are used for therapeutic and/or preventive purposes, different mechanisms could be involved. Interestingly, in some cases, MVs have shown beneficial effects while complete bacterial cells have failed, most probably since MVs are able to penetrate the intestinal epithelial barrier and migrate to other organs [48]. Accordingly, most of the studies describe beneficial effects at the intestinal and immunological level, but more and more information is emerging about beneficial effects observed in pathologies of the skin, nervous system and defence against pathogens, among others, where MVs could be the key players.

## 3. Preventive and/or Therapeutic Application of *Lactobacillus* MVs

Among the mechanisms involved in the beneficial properties exerted by the probiotics, the modulation of the immune response seems to play an important role. In fact, the immunomodulatory effects of probiotics from *Lactobacillus* genus have been extensively studied [49,50], which can be attributed, at least in part, to the production and release of MVs [11].

The immunomodulatory effects of MVs-produced from *Lactobacillus* strains are related to the ability of their cargos to interact with different immune cells located in the gut, such as intestinal epithelial cells (IECs), monocytes and macrophages, lymphocytes, and dendritic cells (DCs). Of note, it has been reported that DCs are key players in the mechanism of action of MVs. DCs are the most available host immune cells to interact with probiotics and/or their vesicles since they are located within the gut-associated lymphoid tissues (GALT) or diffusely distributed throughout the intestinal lamina propria [50]. Furthermore, these are the primary cell type involved in the recognition of a broad spectrum of highly conserved microbial structures through their pattern recognition receptors (PRRs) including Toll-like receptors (TLRs) followed by cytokine production [51]. This type of recognition has been considered essential for the immunomodulatory effects of MVs [11,52,53,54]. In this context, MVs might enter both phagocytic and non-phagocytic cells. Their endocytic entry in non-phagocytic cells occurs through their fusion with lipid rafts, micropinocytosis, clathrin or caveolin-mediated endocytosis or following their interaction with host receptors such as TLR2 [52].

It is interesting to note the ability of some MVs to induce the regulatory CD4 + 25 + Foxp3 + T cells (Treg). Thus, the ingestion of MVs derived from *Lc. rhamnosus* JB-1 (LrJB1-MVs) promoted tolerogenic DCs and, consequently, an increase in Treg lymphocytes in experimental models in mice. These tolerogenic effects were linked to the interaction of MVs with C-type lectin receptors (Dectin-1 and SIGNR1), as well as TLR2 and TLR9 in dendritic cells [55]. On the other hand, the extracellular MVs from *Lc. rhamnosus* GG and *Li. reuteri* DSM 17938 have shown to dampen the responses induced by the pro-inflammatory cytokines interferon-γ (IFN-γ) and interleukin(IL)-17A in human T and natural killer (NK) cells from peripheral blood mononuclear cells (PBMCs), with this effect being exerted through a monocyte-dependent activation [56].

Additionally, *Lp. plantarum* APsulloc 331261-derived MVs (LpAPsulloc331261-MVs) have shown to exert beneficial effects in altered cutaneous immunity, particularly on macrophage polarisation. Thus, in vitro studies demonstrated that Lp-MVs incubation promoted the differentiation of human THP-1 monocytes towards an anti-inflammatory M2 phenotype, by modulating the expression of cell-surface markers and cytokines associated with this phenotype. Furthermore, the treatment of Lp-MVs, applied before or after inflammatory M1 macrophage-favouring conditions with IFN-γ and lipopolysaccharide (LPS), could inhibit M1-associated surface markers and HLA-DRα expression. Moreover, LpMV treatment significantly induced the expression of macrophage-characteristic cytokines such as IL-1β, granulocyte macrophage colony-stimulating factor (GM-CSF) and the anti-inflammatory cytokine IL-10 in human skin organ cultures [57].

It has also been reported that the oral supplementation of *Ligilactobacillus animalis* ATCC 35046-MVs exert pro-angiogenic, pro-osteogenic and anti-apoptotic effects in a glucocorticoid-induced osteonecrosis model in mice [58], maybe due to their functional protein content.

### 3.1. Inflammatory Bowel Disease (IBD)

The term inflammatory bowel disease (IBD) refers to chronic inflammatory conditions, such as Crohn’s disease (CD) and ulcerative colitis (UC), that mainly affect the GIT. However, extraintestinal manifestations frequently occur in these patients, and are considered as a systemic disease [59]. In recent decades, the burden of IBD is rising globally [60,61,62], which could be considered relevant, since different studies have reported that IBD patients show an increased risk for developing different types of intestinal (e.g., colorectal cancer) and non-intestinal related cancer [63,64].

Although the aetiology of IBD is not fully known, it has been proposed that the chronic recurrent intestinal inflammation could be associated with an aberrant immune response to gut bacteria [65], in which the existence of an altered intestinal barrier function may play an important role [66,67,68]. In fact, IBD is characterised by a “leaky gut” in which the damage to intestinal epithelial cells compromises its integrity and loosens tight junctions (TJs). This situation facilitates the direct interaction between pro-inflammatory antigenic components from the lumen and the intestinal epithelium, generating an exacerbated immune response [69].

The goal of IBD therapy is to induce and maintain remission, and this has been typically achieved by downregulating the exacerbated immune response with the administration of immunosuppressive and/or anti-inflammatory drugs, such as aminosalicylates, glucocorticoids, azathioprine, and, more recently, with biological therapies, including infliximab or adalimumab [70]. Although most of these strategies have shown efficacy, there are still many patients with a low response and/or important adverse side effects, especially when the GI integrity is compromised [71]. In this context, alternative preventive strategies, such as the use of MVs as postbiotics, which would modulate the gut microbiota and reinforce normal integrity barrier intestinal functions, could be considered as promising alternatives for the management of these inflammatory intestinal conditions. In fact, different in vivo and in vitro studies have reported the ability of MVs from several probiotics to exert beneficial effects in experimental models of intestinal inflammation.

Thus, the oral administration of MVs from *Lc. paracasei* (Lpi-MVs) to DSS colitis mice attenuated the colonic inflammatory process, which was associated with a decrease in the body weight loss and in the disease activity index (DAI), as well as counteracted the reduction in the colon length [72]. These observations were supported by in vitro assays, since Lpi-MVs have also been shown to ameliorate LPS-induced inflammation in HT-29 colonic cells through endoplasmic reticulum (ER) stress activation [72]. Moreover, the Lpi-MVs incubation with LPS-treated HT-29 cells decreased the activation of inflammation-associated proteins, such as cyclooxygenase-2 (COX-2), inducible nitric oxide synthase (iNOS) and nuclear factor-kappa B (NF-κB).

Similarly, MVs from *Lc. rhamnosus* GG (LrGG-MVs) administration prevented colonic tissue damage as well as colon shortening in DSS-induced colitis in mice. This beneficial effect was reported to be mediated through the inhibition of TLR4-NF-κB-NLRP3 axis and, consistently, the pro-inflammatory cytokines (tumoral necrosis factor-α (TNF-α), IL-1β, IL-6, IL-2) levels were also suppressed in the treated colitis mice. When microbiome studies were performed in the intestinal contents, the 16S rRNA sequencing showed that LrGG-MVs administration could reshape the altered microbiota in this experimental colitis model [73]. Moreover, the intestinal anti-inflammatory effect was also observed when MVs from *Lp. plantarum* Q7 (LpQ7-MVs) were administered to DSS-induced colitis mice, evidenced by the amelioration in the colonic histological damage, which was associated with the downregulation in the production and release of pro-inflammatory cytokines (IL-6, IL-1β, IL-2 and TNF-α) [74]. Additionally, the impact of LpQ7-MVs administration on the gut microbiota was evaluated in this experimental model of colitis, revealing an increase in the abundance of certain bacteria with anti-inflammatory properties, such as *Muribaculaceae* and *Bifidobacteria,* whereas the level of others with pro-inflammatory characteristics, such as *Proteobacteria*, was decreased. Therefore, these observations suggest that LpQ7-MVs could alleviate DSS-induced colitis by modulating the gut microbiota [74].

Moreover, kefir is a fermented dairy product produced by a mixture of yeast and lactic acid bacteria, and its consumption has been associated with health benefits including the amelioration of IBD [75,76]. Based on this background, the impact of MVs obtained from three kinds of kefir-derived *Lactobacillus* strains *(Lentilactobacillus kefiri* KCT 3611 (*Le. kefiri* KCT 3611), *Lactobacillus kefiranofaciens* KCT 5075 (*L. kefiranofaciens* KCT 5075), and *Lactobacillus kefirgranum* KCT 5086 (*L. kefirgranum* KCT 5086)) have been explored in in vitro and in vivo studies. Thus, these MVs improved the inflammatory status in TNF-α-stimulated Caco-2 cells, reducing the mRNA levels and secretion of pro-inflammatory cytokines such as IL-8. These effects were probably mediated by the inhibition of the TNF-α pathway, which consequently decreased the phosphorylation of the p65 subunit of NF-kB. These observations were supported when these MVs were administered to mice challenged with TNBS, since the MVs-treated group significantly reduced the body weight loss and the rectal bleeding, as well as enhanced the stool consistency, with these effects being associated with a reduction in the myeloperoxidase level in serum. In addition, the histological analysis of the colon samples demonstrated that MVs treatment reduced the infiltration of transmural leukocytes and preserved goblet cells [77]. Moreover, additional in vitro and in vivo studies evaluated the effects of *L. kefirgranum* PRCC1301-derived MVs (LkPRCC1301-MVs) on intestinal inflammation and intestinal barrier function. The pre-treatment with LkPRCC1301-MVs in DSS-stimulated Caco-2 cells showed their ability to inhibit the expression of pro-inflammatory cytokines such as IL-2, IL-8, and TNF-α. Furthermore, and evidenced by immunofluorescence analysis, an improvement in the intestinal cell integrity was observed through the recovery of TJ proteins including Zonulin-1 (ZO-1), claudin-1 and occludin. In addition, the administration of LkPRCC1301-MVs to mice submitted to acute (DSS-induced) or chronic (piroxicam-treated IL-10^−/−^) colitis was able to attenuate body weight loss, colon shortening and histological damage in both models. Moreover, the immunohistochemical analysis revealed that phosphorylated NF-κB p65 and nuclear factor of kappa light polypeptide gene enhancer in B-cells inhibitor-α (IκBα) were reduced in the colon tissue sections from colitis mice treated with LkPRCC1301-MVs. These results suggest that LkPRCC1301-MVs might have an anti-inflammatory effect on colitis via the inhibition of the NF-κB pathway in association with an improvement in the intestinal barrier function [78].

Of note, a proteomic analysis performed with MVs from *Lc. casei* BL23 (LciBL23-MVs) revealed that these MVs contain antimicrobial peptides such as p40 and p75 proteins [79], which have been described as anti-apoptotic and anti-inflammatory agents [80]. Interestingly, LciBL23-MVs were able to induce the phosphorylation of epidermal growth factor receptor (EGFR) in the human colon carcinoma cell line T84, showing an effect similar to purified p40 and p75. That suggests that this activity would be mainly due to the presence of these vesicle membrane-bound proteins [81]. Consistently, it has been reported that p40 and p75 regulate IECs antiapoptotic and proliferation responses [80], IECs form the mucosal barrier and protect the host tissue from damaging agents such as luminal pathogens and toxic products, especially through the production and secretion of antimicrobial peptides and chemokines. This may be mediated by the activation of the EGFR/protein kinase B (Akt) pathway [82]. Moreover, p40 has been shown to promote intestinal development in early years, and activate IECs mucin synthesis, contributing to the homeostasis of the intestinal barrier. This supports the fact that p40 has also shown beneficial effects on DSS-induced experimental colitis [83,84,85,86].

IECs are also able to express TLR genes [87], which have a regulating function on the immune system [88], known to be closely related to the gut microbiome composition. In fact, it has been proposed that IBD is associated with an excessive activation of TLR in IECs [89]. Thus, *Lc. casei* ATCC 393-derived MVs (LciATCC393-MVs) have been evaluated in the human Caco-2 cell line, and the results obtained showed that LciATCC393-MVs could slightly increase TLR9 gene expression, as determined by qRT-PCR. Furthermore, the administration of these MVs significantly increased the levels of the anti-inflammatory cytokines IL-4 and IL-10 and, consequently, decreased the levels of the pro-inflammatory cytokines IL-17A and IFN-γ [90]. Similarly, MVs derived from *Latilactobacillus sakei* NBRC 15893 have been described to enhance immunoglobulin A (IgA) production in Peyer’s patches (PPs) murine cells [91]. This effect seems to be exerted through the stimulated production of retinoic acid, nitric oxide and pro-inflammatory cytokines including IL-6, IL-12 and TNF-α, via TLR-2 activation as evidenced in vitro in murine bone marrow-derived DCs (BMDCs) [92].

Finally, *Li. reuteri* BBC3 MVs (LrBCC3-MVs) have also been suggested to show immunomodulatory properties with beneficial effects against gut pro-inflammatory conditions, as evidenced in a chicken model of LPS-induced intestinal inflammation. Thus, the in vitro pre-treatment of LPS-activated chicken macrophages with LrBCC3-MVs downregulated the gene expression of pro-inflammatory cytokines such as *TNF-α*, *IL-1β* and *IL-6*, via the suppression of NF-κB activity, and enhanced the gene expression of IL-10 and TGF-β. These results were confirmed in vivo, since LrBCC3-MVs administration to broilers attenuated the LPS-induced inflammation, by suppressing the LPS-induced expression of pro-inflammatory genes (TNF-α, IL-1β, IL-6, IL-17 and IL-8), and improving the expression of anti-inflammatory genes (IL-10 and TGF-β) [93].

### 3.2. Infectious Diseases

The role that bacterial vesicles can play in different infectious diseases, due to their important functions in intercellular communication and regulation, is well-known. In fact, during an infection process, these vesicles can transfer pathogen mediators that serve as antigens and/or activators of the immune receptors to trigger the host immune response. For instance, the exposure of foetal–maternal structures to *Streptococcus agalactiae A909*-derived MVs can lead to foetal compromise and preterm birth, as evidenced experimentally in mice [94]. Furthermore, MVs produced by pathogenic bacteria, such as *Staphylococcus aureus*, *Streptococcus pneumoniae*, *Bacillus anthracis*, *Streptococcus pyogenes* and *Streptococcus agalactiae* have been reported to include hemolysins and/or pore-forming toxins [94,95,96,97,98,99]. In contrast, it has been reported that MVs derived from *Lactobacillus* species are able to improve the protection against several infectious processes, through different mechanisms, including immunomodulation.

Atopic dermatitis (AD) is a chronic recurrent inflammatory skin disease characterised by itching and xerosis. The pathogenesis of this disease principally involves systemic and local factors, being significantly influenced by the microbial environment and its ability to produce MVs. In fact, these MVs have been involved in the control of the allergic inflammation process and the systemic dysregulation of the immune system that typically occurs in this condition. Thus, the metagenomic analysis of 16s ribosomal DNA extracted from the MVs from healthy control subjects and AD patients showed that levels of *Lactococcus*, *Leuconostoc* or *Lactobacillus* MVs were significantly increased in the control group in comparison with those found in patients with AD. Additionally, in a *S. aureus*-induced experimental model of AD in mice, the administration of *Lp. plantarum* CJLP55-MVs (LpCJLP55-MVs) reduced the epidermal thickening and the levels of the cytokine IL-4. Moreover, in vitro assays supported these results since the lower cell viability induced by *S. aureus*-derived MVs in immortalised human epidermal keratinocytes (HaCaT cells) and macrophages was restored when these cells were pre-treated with LpCJLP55-MVs. All of these data suggest that LpCJLP55-MVs-MVs could help prevent the bacterial-related skin inflammation in AD [100].

*Lp. plantarum* WCFS1-derived MVs (LpWCFS1-MVs) has shown to modulate host responses to vancomycin-resistant *Enterococcus faecium* (VRE) in both *Caenorhabditis elegans* and human colon-derived Caco-2 cells. LpWCFS1-MVs significantly prolonged *C. elegans* survival against VRE infection through the upregulation of the expression of the host defence genes *cpr*-1 and *clec*-60. Furthermore, LpWCFS1-MVs uptake by Caco-2 cells was associated with a significant upregulation of *CTSB* (Cathepsin B), a human homologous gene of *cpr*-1, and *REG3G* (Regenerating islet-derived protein 3-gamma), a human gene that has similar functions to *clec*-60 [101].

Finally, the presence of *Lactobacillus* spp. in the vaginal microbiota is known to play a key role in preventing HIV-1 transmission. In this context, ex vivo studies have reported that the MVs released by *L. crispatus* BC3 and *L. gasseri* BC12, which were isolated from vaginas of healthy women, were able to protect isolated cells and tissues previously infected with HIV-1. The protection exerted by these postbiotics seemed to be related to a partial inhibition of viral attachment to target cells and its entry. This effect was the result of a reduced exposure of envelope glycoproteins (implicated in virus-cell interactions) by MVs-treated HIV-1 virions [102].

### 3.3. Neurological Disorders

The discovery of the microbiota-gut-brain axis has been a revolution in the understanding of systemic influences on brain function. It refers to a wide range of interactions between the gut microbiota and the central nervous system involving immune, endocrine, and neural signalling pathways [103]. Sudo et al. (2004) were one of the first to report the influence of gut microbiota in brain function [104], showing that germ-free mice have a hyperactive hypothalamic–pituitary–adrenal (HPA) axis with higher levels of stress-associated hormones after restraint stress than those mice with conventional microbiota [104]. Since then, numerous studies have described the role played by gut microbiota in the modulation of the central and enteric nervous system, stress responses, anxiety [105,106,107] and memory [103].

In fact, several bacteria have been shown to produce different neurotransmitters such as serotonin, dopamine, gamma-aminobutyric acid (GABA) or norepinephrine [108]. Specifically, some *Lactobacillus* probiotic species have been identified to produce a wide range of neurotransmitters and associated molecules implicated in its antidepressant properties [109]. For example, *Lactobacillus helveticus* NS8 and MCC1848 intake enabled the recovery of chronic and subchronic-stressed rodents from their state of depression through the modulation of the central 5-Hydroxytryptamine (5-HT) system and Brain Derived Neurotrophic Factor (BDNF) expression [110,111]. Similarly, *Lp. plantarum* MTCC 9510 supplementation has been described to reduce monoamine oxidases (MAOs) levels and the associated oxidative stress in brain tissues of mice, thus preventing stress-induced behavioural alteration (depression, anxiety, learning and memory, stereotypic behaviour) [112]. Moreover, in another study, *Lp. plantarum* PS128 was able to increase dopamine levels in the striatum of mice, which resulted in amelioration in anxiety-like behaviours [113] and suggests the ability of some strains of *Lp. plantarum* to modulate the central dopamine system, with beneficial effects for mood disorders.

Of note, bacterial EVs seem to be also involved in this intercommunication due to their capacity of passing through the TJs of the intestinal wall, and into the bloodstream [114,115]. This has been evidenced by the presence of bacterial RNA in the blood of healthy individuals, which would be metabolised without protection in the circulation [116,117,118]. Afterwards, they could be distributed throughout the body, and deliver the cargo to several organs including the brain. In fact, the studies performed by Zakharzhevskaya et al. [119] revealed that GABA, the major inhibitory neurotransmitter, and its intermediate metabolites were detected in OMVs. Additionally, other mediators such as histamine were also detected, which are clearly involved in the regulation of intestinal function and modulation of local immune responses, being also able to exert actions on diverse brain functions [120]. These findings confirm the capacity of bacteria, commensal or probiotics, to package and excrete active molecules in protected EVs and shuttle it through the body at biologically active levels.

Supporting this, different in vitro studies have reported the ability of LpCJLP55-MVs to upregulate the expression of BDNF transcripts, as well as the proBDNF protein, in HT22 hippocampal cells after the induction of depression-associated changes by corticosterone administration [121]. It was proposed that the presence of sirtuin 1 (Sirt1) in these MVs could contribute to the observed effects on BDNF expression. Sirt1 is a deacetylase that contributes to cellular regulation in response to stress. This was supported by an in vivo study using mice submitted to restraint stress to generate a depressive phenotype. LpCJLP55-MVs were injected intraperitoneally either during restraint stress, immediately following restraint stress or 2 weeks following stress exposure, and the treatments normalised BDNF expression and stress-induced behaviours, similarly to the effects obtained with the antidepressant drug imipramine [121]. More recently, it has been noted that LpCJLP55-MVs treatment in HT22 cells reversed glucocorticoid (GC)-induced reduced expression of BDNF, Nt4/5 and Sirt1 expression, as well as that of Mecp2, another epigenetic factor that regulates BDNF and Nt4/5 expression, although this was only achieved partially [122]. All this data suggests that the MVs released by *Lp. plantarum* CJLP55 plays a significant role in modulating the expression of neurotrophic factors in the hippocampus, thus affording antidepressant-like effects in mice with stress-induced depression.

Of note, some evidence indicates the ability of EVs from commensal bacteria to interact with the peripheral nervous system, although this knowledge is still limited. Thus, the presence of LrJB1-MVs in the intestinal lumen have been reported to result in a high excitability of afferent neurons in the myenteric plexus in mice, as evidenced by the increased number of action potentials recorded in adjacent patch-clamped sensory neurons [55].

Additionally, the interaction of MVs with the enteric nervous system can generate changes in peristaltism and local movements of the gut. In fact, ex vivo experimental models of peristalsis have shown the ability of LrJB1-MVs to generate changes in nerve-dependent colon migrating motor complexes (MMCs) [55]. Moreover, *Li. reuteri* DSM 17938-MVs decreased the velocity and frequency of propagating contractile cluster contractions in the jejunum and increased them in the colon as gut bacteria contribute to gut motility [123].

### 3.4. Cancer

The potential beneficial effects of probiotics against cancer have been supported by different in vitro and in vivo studies [124]. Specifically, several *Lactobacillus* species such as *L. acidophilus*, *Lc. casei* and *Lc. rhamnosus* GG have been described as exerting protective effects in experimental models of intestinal cancer in rats [125,126,127]. Although the mechanisms involved in these effects are not fully understood, their ability to improve host responses to tumours, as well as the regulation of apoptosis and cell differentiation [128], has been proposed, which could involve the participation of their MVs.

In this sense, LrGG-MVs have shown antiproliferative effects in colorectal cancer cells, SW480 and HT-29 cell lines, most probably by modulating the expression and production of the carcinoembryonic antigen (CEA) gene [129]. Moreover, these MVs have shown to inhibit cell proliferation in the hepatic cancer cells HepG2, which seems to be mediated by the increase in *bax*/*bcl*-2 ratio that leads to cancer cell death [130].

Although, nowadays, few studies provide evidence of the beneficial effects of the MVs on cancer therapy, the findings above provide new perspectives and ideas in this field as well as establish the potential use of probiotic MVs for the prevention of different types of cancer.

## 4. Conclusions

*Lactobacillaceae*-derived MVs have been shown to promote beneficial effects for the prevention and treatment of a wide range of diseases. Several studies have shown promising results in the application of MVs on host health modulation and in pathological conditions, especially in IBD and related affections (Table 1).

These findings may suggest the therapeutic potential that MVs released by *Lactobacillaceae* strains could perform as an alternative to the utilisation of probiotics. This possibility is mainly enabled by the safe nature of MVs in comparison to the risks that the consumption of the entire bacteria could entail in certain cases. However, although the immunomodulatory and anti-inflammatory properties of MVs have been widely demonstrated in preclinical studies, this clinical use has not been explored yet. Consequently, further investigation is needed to confirm the application of Lactobacillus-derived MVs as a new therapeutic strategy in human health.

## Figures and Tables

**Figure 1 nutrients-14-05296-f001:**
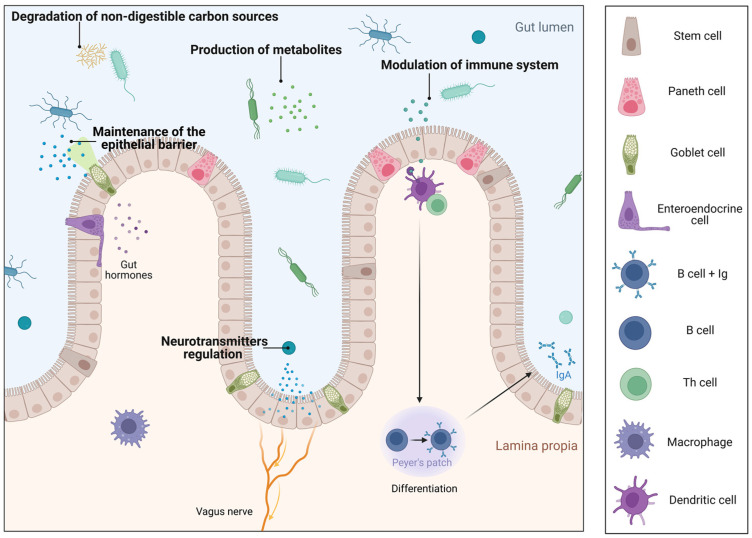
Specific gut microbiota functions. Gut microbiota impacts on host nutrient metabolism through the degradation of non-digestible carbon sources and the synthesis of metabolites, mainly SCFAs and vitamins. Additionally, it plays an essential role in the protection against pathogens in two ways, firstly by its participation in the maintenance of the structural integrity of the gut mucosal barrier. Secondly, through the modulation of the immune system, it stimulates lymphocyte B differentiation and the consequent release of IgA to the gut lumen. Finally, microbiota can also participate in neurotransmitter regulation and exert a beneficial effect on the nervous system. Created with BioRender.com. Accessed on 1 October 2022.

**Figure 2 nutrients-14-05296-f002:**
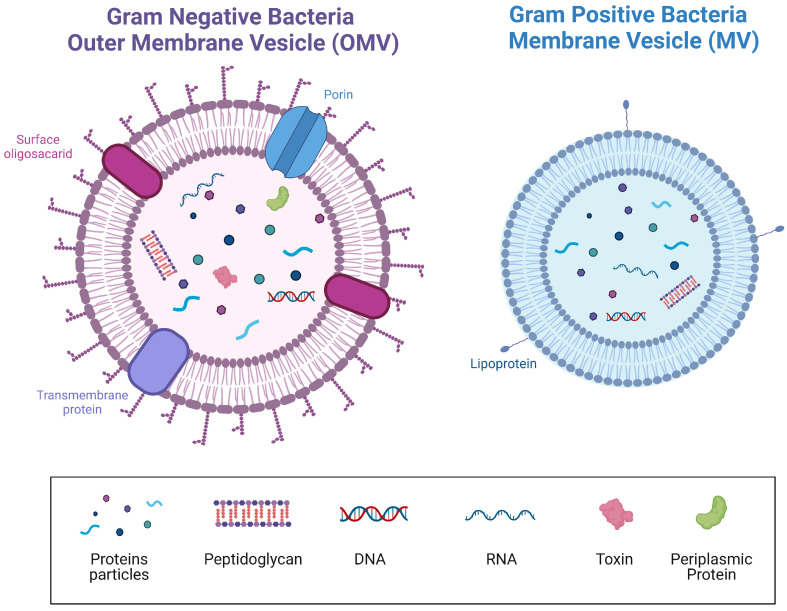
Bacterial extracellular membrane vesicles (EVs). Structure and cargo types comparing Gram-negative and positive bacteria. Created with BioRender.com. Accessed on 1 November 2022.

**Table 1 nutrients-14-05296-t001:** Effects of different Lactobacillus strains in pre- and clinical studies.

Specie	Indication	Model	Mechanism	Ref.
*L. animalis* ATCC 35046	Osteonecrosis	Glucocorticoid- induced osteonecrosis mice	↑ Angiogenesis and osteogenesis. ↓ Cell apoptosis.	[58]
*Lc. casei* BL23	IBD	T84 cell line	Anti-apoptotic and anti-inflammatory effects exerted by p40 and p75 proteins.	[81]
*Lc. casei* ATCC 393	Colitis	Caco-2 cell line	↑ Anti-inflammatory mediators (TLR9 gene expression and levels of IL-4 and IL-10). ↓ Pro-inflammatory markers (IL-17A and IFN-γ).	[90]
*L.crispatus* BC3 and *L. gasseri* BC12	HIV infection	TZM-bl and MT-4 cells	Partial inhibition of viral attachment to target HIV cells and its entry.	[102]
*Le. kefiri* KCT 3611, *L. kefiranofaciens* KCT 5075, and *L. kefirgranum* KCT 5086	Colitis	Caco-2 cell line	↓ Inflammatory process: ↓ TNF-α pathway. ↓ p65 phosphorylation.	[77]
		TNBS-IBD-induced mice	↓ Body weight loss and rectal bleeding. ↓ Infiltration of transmural leukocytes, goblet cells and seric levels of myeloperoxidase. ↑ Stool consistency.	
*L. kefirgranum* PRCC-1301	Colitis	DSS-stimulated-Caco-2 cells	↑ Intestinal cell integrity: recovery of TJs proteins ↓ Pro-inflammatory cytokines (IL-2, IL-8, and TNF-α).	[78]
		DSS-induced colitis mice	↓ Body weight loss, colon shortening, and histological damage. ↓ Phosphorylation of NF-κB p65 and IκBα in colon tissue.	
Specie	Indication	Model	Mechanism	Ref.
*Lc. paracasei*	IBD	HT-29 cell line	Anti-inflammatory effect: ↓ Inflammation-associated proteins (COX-2, iNOS, NFκB, and NO).	[72]
		DSS-induced colitis mice	↓ Weight loss and DAI. Maintenance of colon length.	
*Lp.* *plantarum* APsulloc 331261	Skin inflammation	THP-1 cells	↑ Anti-inflammatory M2 phenotype. ↓ M1-associated surface markers and HLA-DRα expression in pro-inflammatory M1 macrophage-favouring conditions.	[57]
		Human skin organ cultures	↑ IL-1β, GM-CSF and IL-10.	
*Lp.* *plantarum* Q7	Colitis	DSS-induced colitis mice	↓ Histological damage. ↓ Pro-inflammatory cytokines. -Modulation of gut microbiota: ↑ Anti-inflammatory bacteria (Muribaculaceae and Bifidobacteria). ↓ Pro-inflammatory bacteria (Proteobacteria).	[74]
*Lp. plantarum* CJLP55	Atopic dermatitis	HaCaT cells and macrophages treated with S. aureus MVs	Restoration of cell viability.	[100]
		Human clinical trial	↑ Proportion of Lactobacillus MVs in the control group. ↓ Epidermal thickening and cytokine IL-4 levels in AD patients.	
	Depression	HT22 hippocampal cells	↑ Expression of BDNF and proBDNF protein. ↑ BDNF regulating factors (Nt4/5 and Mecp2).	[121]
		Stress-induced depression mice	Normalisation of BDNF expression.	[122]
Specie	Indication	Model	Mechanism	Ref.
*Lp. plantarum* WCFS1	VRE infection	Caco-2 cell line	Modulation of host response. ↑ Expression of host defence genes (CTSB and REG3G).	[101]
		C. elegans	↑ Expression of host defence genes (Cpr-1 and clec-60).	
*Lc.* *rhamnosus* JB-1	Immune system	PP-derived DCs	Activation of tolerogenic dendritic cells. ↑ Treg cells.	[55]
	Brain function	Ex vivo mice model of peak pressure-induced MMC in segments of colon	↓ Excitability of afferent neurons in the myenteric plexus.	[55]
	Enteric nervous system	Ex vivo mice model of peristalsis	↓ Amplitude of neuronally dependent MMCs.	[55]
*Lc.* *rhamnosus* GG	Immune system	PBMCs-derived T and NK cells.	↓ Pro-inflammatory cytokines (IFN-γ and IL-17A).	[56]
	Colitis	DSS-induced colitis mice model.	↓ Colonic tissue damage and colon shortening. Reshape of the gut altered microbiota. ↓ Pro-inflammatory cytokines (TNF-α, IL-1β, IL-6, IL-2).	[73]
	Colorectal cancer	SW480 and HT-29 cell lines	Anti-proliferative effect: ↑ Gene expression and protein synthesis of CEA.	[129]
	Hepatic cancer	HepG2 cell line	Antiproliferative effect: ↑ Bax/Bcl-2 ratio.	[130]
*Li. reuteri* BBC3	IBD	LPS-activated chicken macrophages	↓ Pro-inflammatory cytokines (TNF-α, IL-1β and IL-6) via the suppression of NF-κB activity.	[93]
		LPS-induced intestinal inflammation in broilers	↑ Growth performance. ↓ Intestinal injury and mortality. Anti-inflammatory function: ↓ Pro-inflammatory genes (TNF-α, IL-1β, IL-6, IL-17 and IL-8). ↑ Anti-inflammatory genes (IL-10 and TGF-β).	
Specie	Indication	Model	Mechanism	Ref.
*Li. reuteri* DSM 17938	Enteric nervous system	Ex vivo model: mouse jejunal and colonic segments	Modulation of velocity and frequency of propagating contractile cluster contractions: ↑ Colon ↓ Jejunum	[123]
	Immune system	T and NK cells from PBMCs	↓ Pro-inflammatory cytokines (IFN-γ and IL-17A).	[56]
*L. sakei* NBRC 15893	Immune system	Murine bone marrow-derived DCs and murine PP cells	↑ IgA production in PP cells. ↑ Gene expression of iNOs, RA, and pro-inflammatory cytokines 8IL-6, IL-10, IL-12 and TNF-α).	[91,92]

Abbreviation list: AD: atopic dermatitis, BDNF: brain-derived neurotrophic factor, CEA: carcinoembryonic antigen, COX-2: cyclooxygenase-2, DAI: disease activity index, DCs: dendritic cells, DSS: dextran sodium sulfate, GM-CSF: granulocyte-macrophage colony-stimulating, HIV: human immunodeficiency virus, IBD: inflammatory bowel disease, IκBα: nuclear factor of kappa light polypeptide gene enhancer in B-cells inhibitor-α, IL: interleukin, INF-γ: interferon-γ, iNOS: inducible nitric oxide synthase, LPS: lipopolysaccharide, MMCs: migrating motor complexes, MVs: microvesicles, NFκB: nuclear factor κB, NK: natural killer, NO: nitric oxide, PBMCs: peripheral blood mononuclear cells, PP: peyer’s patch, proBDNF: pro brain-derived neurotrophic factor, RA: retinoic acid, TGF-β: transforming growth factor β, TNBS: 2,4,6-trinitrobenzene sulfonic acid, TNF-α: tumor necrosis-α, TLR9: toll-like receptor 9. Table symbols: ↓ Reduction and ↑ increase.

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
