# Peer review of "Novel Horizons in Postbiotics: *Lactobacillaceae* Extracellular Vesicles and Their Applications in Health and Disease"

_nutrients, 2022, doi:10.3390/nu14245296_

Round 1

Reviewer 1 Report

The manuscript entitled "Novel horizons in postbiotics: Lactobacillaceae extracellular vesicles and their applications in health and disease" is interesting, and can give us a further understanding the Lactobacillaceae extracellular vesicles' function. The manuscript is well organized. But the manuscript needs some minor revision before proceeding.

1. Line 31: "1014", or "1014"?

2. Line 379 "ex vivo" shoud be in italic;

3. The references format should be revised carefully.

Author Response

The manuscript entitled "Novel horizons in postbiotics: Lactobacillaceae extracellular vesicles and their applications in health and disease" is interesting, and can give us a further understanding the Lactobacillaceae extracellular vesicles' function. The manuscript is well organized. But the manuscript needs some minor revision before proceeding.

  1. Line 31: "1014", or "1014"?
  2. Line 379 "ex vivo" shoud be in italic;
  3. The references format should be revised carefully.

Response: We appreciate the reviewer suggestion and all reviewer comments have been considered. Following all considerations, the paper has been revised.

Reviewer 2 Report

The authors conducted a thorough review of the current knowledge about extracellular vesicles from Lactobacillus strains, particularly their effects in host health and various disease models and conditions. The manuscript is very well written. I only have two minor comments:

1. can the authors make sure that they make a footnote under Table 1 to spell out the abbreviations used in the table. 

2. The figure legend for Fig 1 can be enhanced. The authors should describe what the figure shows in a bit more detail. 

Author Response

The authors conducted a thorough review of the current knowledge about extracellular vesicles from Lactobacillus strains, particularly their effects in host health and various disease models and conditions. The manuscript is very well written. I only have two minor comments:

  1. can the authors make sure that they make a footnote under Table 1 to spell out the abbreviations used in the table. 

Response: We greatly appreciate the reviewer's comment.  An abbreviation list used in Table 1 has been included.

  1. The figure legend for Fig 1 can be enhanced. The authors should describe what the figure shows in a bit more detail. 

Response: Following the reviewer's suggestion, a detailed figure legend for Figure 1 has been added.